# Influence of a Soft Story on the Seismic Response of Non-Structural Components

Vyshnavi Pesaralanka [1], S. P. Challagulla [1,*], Felipe Vicencio [2], P. Suresh Chandra Babu [3], Ismail Hossain [4], Mohammed Jameel [5] and Uppari Ramakrishna [6]

1    Department of Civil Engineering, Koneru Lakshmaiah Education Foundation, Vaddeswaram, Guntur 522302, Andhra Pradesh, India
2    Facultad de Ingenieria, Arquitectura y Diseño, Universidad San Sebastian, Santiago 8420524, Chile
3    Department of Civil Engineering, Malla Reddy Engineering College, Maisammaguda, Hyderabad 500100, Telangana, India
4    School of Natural Sciences and Mathematics, Ural Federal University, 620000 Yekaterinburg, Russia
5    Department of Civil Engineering, College of Engineering, King Khalid University, Asir, Abha 61421, Saudi Arabia
6    Department of Civil Engineering, Vignana Bharathi Institute of Technology, Hyderabad 501301, Telangana, India
*    Correspondence: chsuryaprakash@kluniversity.in

**Abstract:** Multi-story, reinforced-concrete (RC) building structures with soft stories are highly vulnerable to damage due to earthquake loads. The soft story causes a significant stiffness irregularity, which has led to numerous buildings collapsing in previous seismic events. In addition to the structural collapse, the failure of non-structural components (NSCs) has also been observed during past earthquakes. In light of this, this study investigates the effect of a soft story and its location on the seismic behavior of a supporting building and NSCs. The soft story is assumed to be located on the bottom (ground), middle, and top-story levels of the considered building models. Story displacements and inter-story drift ratios are evaluated to assess structural behavior. The floor response spectra and the amplification effects of NSC on the floor acceleration responses are studied to understand the behavior of NSCs. The analysis results revealed that the bottom soft story exhibits a considerable vertical stiffness irregularity, and its position substantially affects the floor response spectra. The amplification in the floor acceleration response was found to be greater at the soft-story level. This study reported that middle soft-story buildings exhibit the most remarkable amplification in the component's acceleration. Finally, peak floor response demands are compared with the code-based formulation, and it is found that the code-based formulation's linear assumption may lead peak floor response demands to be underestimated or overestimated.

**Keywords:** floor amplification factor; non-structural component; soft story; inter-story drift ratio; story displacement

## 1. Introduction

Non-structural components (NSCs) and elements of buildings do not resist loads [1]. Based on the types of failure, NSCs can be classified as acceleration-sensitive components and displacement/drift-sensitive components [2]. Even though precise methods for reliably estimating seismic demand on both acceleration and displacement-sensitive non-structural components are now available, simpler procedures are sometimes required in design scenarios [3]. National and international codes give several simpler formulas for calculating seismic demand on NSEs. The fundamental goal of many seismic codes in use in earthquake-prone regions is to estimate the maximum acceleration, and hence the maximum inertial force, caused by the predicted seismic shaking on NSC. As a result, the current study is limited to the acceleration-sensitive, non-structural components. Acceleration

failure is due to the inertial forces produced in the component. Suspended building utility systems, such as pipe systems and cable trays, and anchored or free-standing building utility systems or contents are examples of acceleration-sensitive, non-structural components. Damage to NSCs might result in more considerable direct and indirect economic losses than principal structural members. The destruction of NSCs, including essential and expensive equipment, may affect the functioning of structures, particularly critical facilities such as hospitals, airports, and historic or culturally valuable systems [4,5]. These findings demonstrate that the seismic performances of NSCs are as important as those of structural components. The present Standards and Guidelines have been created mainly based on empirical techniques developed from prior experiences and engineering expertise [6]. Thus, non-structural components must be earthquake-designed to keep them safe and ensure that the building can continue to function after an earthquake. In order to accomplish this, the floor response spectrum (FRS) needs to be determined at the point where the non-structural component is attached to the primary structure.

The floor response spectrum (FRS) approach is a decoupled analysis method [7–10]. The primary structure is dynamically analyzed first, without regard for the influence of the secondary system. At the floor level where a NSC is attached, the acceleration response history is used as input to a secondary structure to construct the FRS. Therefore, the maximum design force for the design of the NSCs can be obtained from the generated FRS. The seismic performance of components exposed to the ground motion was studied, and it was concluded that the amplification in the response of the primary structure would increase the damage probability of NSCs [11]. In the 1970s, researchers started looking into FRS generation techniques. The NSC and its supporting structure were formerly typically treated as single degree-of-freedom (SDOF) systems in several methods. Yasui et al. [12] developed a method for generating the smooth design floor response spectra utilizing the design spectra or ground response spectra. A novel technique is developed and validated for directly determining floor acceleration spectra [13]. Wei Jiang et al. [14] constructed floor response spectra to examine seismic demands on nuclear plants and concluded that the FRS from time history analysis had considerable variations, particularly in tuning cases. Very recently, Ruggieri and Vukobratovic [15] studied the effect of flexible diaphragms on the peak floor accelerations (PFAs) and floor acceleration spectra of single-story buildings. The analysis concluded that the flexible diaphragm significantly affects FRS and PFAs. The floor response spectrum of multi-storied structures [16–20] has been investigated. Little research has considered the nonlinear behavior of NSCs. Vukobratovic and Fajfar [21] proposed the code-oriented method for the determination of FRS, and it was shown that the non-linear behavior of NSCs reduces the FRS values. Inelastic floor acceleration spectra were developed for the design of acceleration-sensitive NSCs [22]. Vukobratovic and Ruggieri [23] recently investigated the floor acceleration demands in a twelve-story, reinforced-concrete shear wall building, and the analysis concluded that even with a modest ductility demand of 1.5, the nonlinear behavior of non-structural components resulted in a favorable decrease in floor response spectra, notably in the resonance areas. Little recent research has examined the impact of structural irregularities on the FRS. The effect of a vertical stiffness irregularity on the floor response spectrum was investigated [24], and the research shows that the amplification of the floor acceleration is higher at the soft-story level. The impact of a torsional irregularity on the tri-directional response spectra of the industrial buildings was studied [25], and the analysis concluded that the buildings with variations in mass and stiffness result in FRS intensification. Although numerous FRS generation techniques have been documented in the pertinent literature [14,19,26,27], none adequately examine the effect of a vertical stiffness irregularity (soft story) present at different floor levels on the seismic behavior of the non-structural components.

A considerable number of moment-resisting frame structures have been built all over the world. These structures require open stories (or soft stories) for parking garages, reception lobbies, retail shops, and meeting rooms. The presence of a soft story induces structural irregularity, and structures with such irregularity are highly sensitive to damage

under earthquake loads. A structure with a soft story has a stiffness discontinuity due to the open story's high flexibility compared to the adjacent stories. The stiffness discontinuity in a story is the one with less lateral stiffness than the story above [28,29]. Many researchers have conducted studies to explore the seismic performance of soft-story buildings. Saraswati and Vineet [30] investigated a bare frame structure with a soft story and concluded that the presence of a soft story in a structure reduced base shear and enhanced the rapid shift in drift. The story displacement and moments are significantly affected by the variations of the soft story level under near-fault ground motions, and the level of the soft story has a significant effect on the story shear forces [31]. Alam and Amanat [32] studied infilled reinforced-concrete frames. They concluded that the soft-ground storied buildings have a longer period of vibration, and the drift of the open-ground columns is much higher in the presence of infills on the upper floors. Das and Nau [33] investigated the inelastic seismic response of multistory structures with stiffness irregularity. They observed that sudden changes in seismic response occurred around the presence of irregularities. The presence of irregularities in the lower floors caused the most variability in the seismic response [34]. Choi [35] investigated the effect of vertical mass irregularity on the seismic response of multi-story structures. The study reported that the seismic response was higher when mass irregularity was located on the top floor. The location of an irregularity and the magnitude of an earthquake had the most impact on the seismic response [36]. A few of the latest studies have investigated the location of a vertical stiffness irregularity (soft-story) on the seismic response of the building structures. According to the latest study [37], when a structure is subjected to seismic loading, the stiffness at the base has a substantial influence on the overall stability and response of the structure. Samyak and Debarati [38] studied the seismic response of the stiffness of irregular steel frames under mainshock and aftershock. They considered stiffness irregularity at the building frames' bottom, middle, and top stories. Their analysis concluded that the stiffness irregularity at the bottom story causes a maximum inter-story drift ratio (IDR). In a very recent study [39], the effect of a soft story and its level on the moment-resisting frame (MRF) is investigated. From the analysis results, the authors concluded that changing the soft-floor level significantly impacts structural response. The majority of research investigations mentioned in this paragraph have been conducted on the soft story in terms of infill walls that can dissipate the energy induced by an earthquake. In the case of bare frames, however, a change in soft-story height significantly influences the stiffness of the building floors. This might substantially impact the building's structural behavior, affecting the floor response spectra.

From the previous studies, it is evident that a vertical stiffness irregularity (soft story) significantly affects the structural and non-structural component's seismic response under earthquake loads. Though a recent study [24] focused on the effect of vertical stiffness irregularity on the floor response spectrum, it was limited to simple 2D frames. The impact of a soft story on the seismic performance of non-structural components requires attention. Hence, this study explores the effect of a soft story and its location on the seismic performance of a building structure and NSCs. Story displacements and inter-story drift ratios are the two response parameters that assess the structural behavior of the building models. The floor amplification factors, peak component acceleration, component dynamic amplification factors, and floor response spectra are essential in assessing seismic demands on NSCs. In the generation of FRS, component dynamic amplification factors play a crucial role, as they reflect the amplification of NSCs. Therefore, all the specified factors and spectra are evaluated for building models under earthquake loads. The amplification factors are compared with those obtained from the code-based formulations. It is important to keep in mind that the recent study [40] emphasized the performance-based seismic design of non-structural components. The accurate estimate of seismic demand in the performance evaluation of NSCs necessitates the adoption of an appropriate EDP. Fragility curves for NSCs are frequently represented in terms of floor spectral acceleration in loss estimate studies [41]. The approach given in FEMA P-58 [42] for estimating damage and loss is time-consuming and expensive, and a new direct loss measure (LM) was developed in the

most recent study by Sani et al. [43]. The Hazus-MH 2 [44] provides four damage states for NSC damage assessment: slight, moderate, extensive, and complete. Damage is classified as slight, moderate, extensive, or complete by the fragility curves. However, the current study did not investigate the damage and loss estimation studies of NSCs.

The organization of the paper can be broken down into the following sections: Section 2 describes the modelling and analysis of considered building models. Section 3 gives the details of ground motion. Section 4 presents the results and discussion, and concise conclusions are drawn in the last section (i.e., Section 5).

## 2. Modelling and Analysis of Buildings

The present study considered a set of reinforced-concrete (RC) buildings with an identical plan. The considered five-story (G + 4) and ten-story (G + 9) 3D structural models are shown in Figure 1. The chosen structures represent the dynamic behavior of medium- and high-rise buildings, respectively [19]. The structural models are considered RC moment-resisting frames (MRF). A story height of 3 m has been kept constant for all the reference building models (without a soft story). In the case of soft-story buildings, two heights (5 m and 7 m) are assumed for a soft-story level. A bay width of 3 m has been kept constant for all the models. In the current study, the five- and ten-story reference buildings are designated $M_{5ref}$ and $M_{10ref}$, respectively. The soft story is assumed to be presented at the building models' bottom, middle, and top levels. In the case of a five-story building, models with a soft story (height = 5 m) at the bottom, middle, and top levels are designated as $M_{5bs5}$, $M_{5ms5}$, and $M_{5ts5}$, respectively, and can be seen in Figure 1b–d. The notations $M_{5bs7}$, $M_{5ms7}$, and $M_{5ts7}$ represent a five-story building with a soft story (height = 7 m) at the bottom, middle, and top levels, respectively. In the case of a ten-story building model, $M_{10bs5}$, $M_{10ms5}$, and $M_{10ts5}$ models have a soft-story height of 5 m on the bottom, sixth, and top floor levels, respectively, and can be seen in Figure 2. Also, the models $M_{10bs7}$, $M_{10ms7}$, and $M_{10ts7}$ have a soft-story height of 7 m. Overall, 14 models were considered in this study. All soft-story building models satisfy the irregularity criteria as per IS 1893 (Part 1) 2016 [28]. The assumed soft-story heights are based on the available literature [24,37,39,45] to fulfill the irregularity criteria. The stiffness irregularity checks ($K_i/K_{i+1}$) are carried out in both plan directions as per IS 1893 standard, and the corresponding values are shown in Table 1.

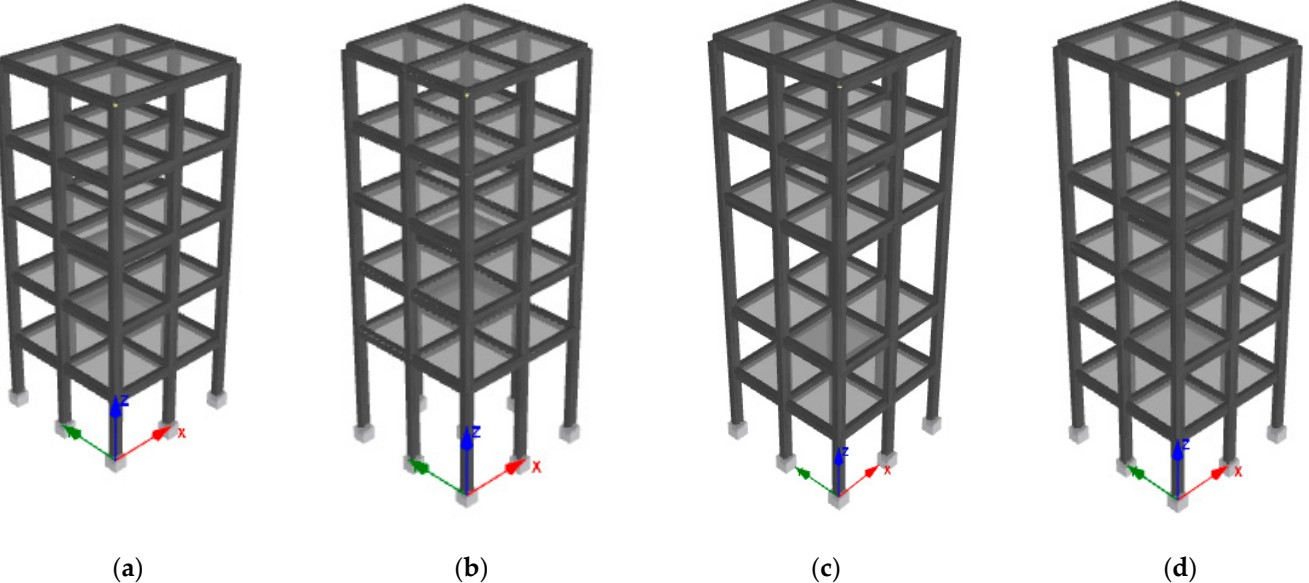

(**a**)    (**b**)    (**c**)    (**d**)

**Figure 1.** Five-story buildings. (**a**) Reference building; (**b**) bottom soft-story building; (**c**) middle soft-story building; (**d**) top soft-story building.

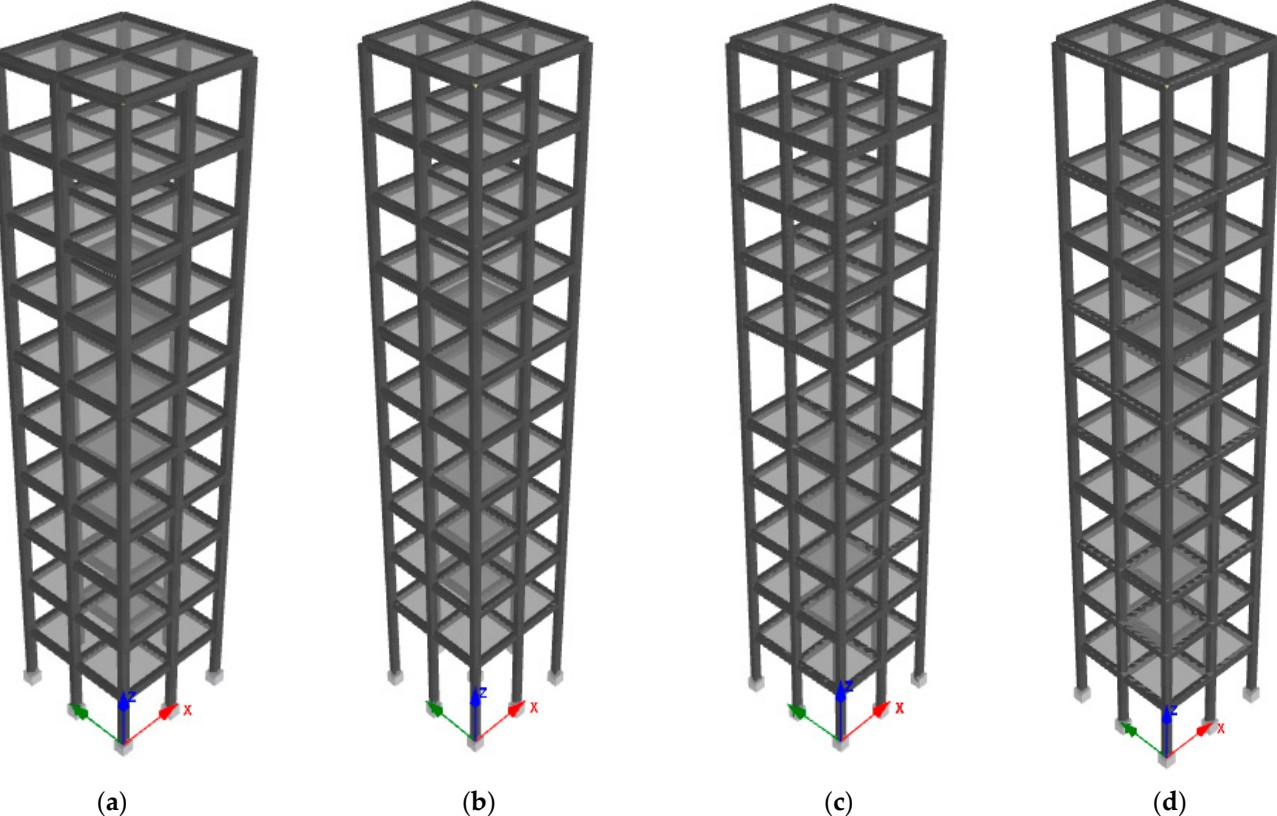

**Figure 2.** Ten-story buildings. (**a**) Reference building; (**b**) bottom soft-story building; (**c**) middle soft-story building; (**d**) top soft-story building.

**Table 1.** Stiffness irregularity ratios of the building models.

| Building Model | Heigh of a Soft Story (m) | Soft Story Location (Floor Level) | $K_i/K_{i+1}$ X-Dir | Y-Dir |
|---|---|---|---|---|
| $M_{5bs5}$ | 5 | 1 | 0.495 | 0.373 |
| $M_{5ms5}$ | 5 | 3 | 0.407 | 0.334 |
| $M_{5ts5}$ | 5 | 5 | 0.348 | 0.302 |
| $M_{10bs5}$ | 5 | 1 | 0.522 | 0.389 |
| $M_{10ms5}$ | 5 | 6 | 0.449 | 0.365 |
| $M_{10ts5}$ | 5 | 10 | 0.330 | 0.305 |
| $M_{5bs7}$ | 7 | 1 | 0.231 | 0.159 |
| $M_{5ms7}$ | 7 | 3 | 0.211 | 0.154 |
| $M_{5ts7}$ | 7 | 5 | 0.173 | 0.134 |
| $M_{10bs7}$ | 7 | 1 | 0.245 | 0.168 |
| $M_{10ms7}$ | 7 | 6 | 0.235 | 0.172 |
| $M_{10ts7}$ | 7 | 10 | 0.188 | 0.151 |

The building models are assumed to be located in the highest seismic zone (Zone V, as per IS 1893–2016). Grades of concrete and steel are taken as M 30 and HYSD 415 for reinforced-concrete modelling. Floor finishes and live load have been set at 1.5 kN/m$^2$ and 3 kN/m$^2$, respectively, as per IS 875-Part 2 [46]. The preliminary dimensions of column and beam have been chosen as per IS 13920: 2016 [47]. The column sizes (300 mm × 450 mm) and beam sizes (230 mm × 450 mm) have been kept uniform for frames. The thickness of the RC slab is set to 150 mm for all frames. Early floor response spectrum techniques were based on the assumption that during earthquakes, buildings and NSCs remain in

the linear elastic zone. Currently, the scope of this work is confined to structural systems that respond elastically. The inelastic behavior of the structure and/or NSC is outside the scope of this study but will be taken into account in future research. The elastic model of the structure employed as a reference case aims to imitate the theoretical behavior of the structures while ignoring nonlinear effects during the dynamic response. To assess the seismic behavior of the building models, the elastic response of bare frames is explored from the bi-directional time-history analysis using the finite element package software SeismoStruct [48]. The elastic frame element is used to simulate the beams and columns. The compression behavior of confined concrete is defined by the model developed by Mander et al. [49]. The tension behavior of steel reinforcement is accounted for using the Menegotto-Pinto [50] steel model. RC slabs are modelled as a rigid diaphragm using the penalty functions nodal constraints technique. Based on the work of Pinho et al. [51], the penalty function exponent for the building is calculated and is $10^{10}$ in this study. A Rayleigh damping model of 5% (associated with the lowest mode and the highest mode, resulting in a total of 95% cumulative mass participation in both directions) is defined to model the damping effects in the dynamic analyses. The ground motion selection approach employed in the current investigation is discussed in the next section.

## 3. Selection and Scaling of Ground Motions

In the seismic response assessment procedure, actual ground-motion records produce a realistic response [52,53]. The Pacific Earthquake Engineering Research Center (PEER) [54] NGA-West2 Database has such records readily available. Hence, in the present study, 11 horizontal ground motion excitations have been considered as per ASCE 7–16 [29] for hard soil type. According to the National Earthquake Hazard Reduction Program (NEHRP) [55] guidelines, ground motions are chosen based on shear wave velocity ($V_{S30}$) to represent hard soil. The details of the excitation are shown in Table 2. Spectrum compatible ground motions are utilized in this study because they can significantly reduce the computing work compared to several ground motions [56]. The time-domain spectral-matching approach suggested by [57] is used to produce spectrum-compatible earthquake excitations.

**Table 2.** Details of ground motions for time history analysis.

| Earthquake | Year | Station | $M_w$ | $R_{jb}$ (km) | $V_{s30}$ (m/s) |
|---|---|---|---|---|---|
| Helena_Montana-01 | 1935 | Carroll College | 6 | 2.07 | 593.35 |
| Helena_Montana-02 | 1935 | Helena Fed Bldg. | 6 | 2.09 | 551.82 |
| Kern County | 1952 | Pasadena—CIT Athenaeum | 7.36 | 122.65 | 415.13 |
| Kern County | 1952 | Santa Barbara Courthouse | 7.36 | 81.3 | 514.99 |
| Kern County | 1952 | Taft Lincoln School | 7.36 | 38.42 | 385.43 |
| Southern Calif | 1952 | San Luis Obispo | 6 | 73.35 | 493.5 |
| Parkfield | 1966 | Cholame—Shandon Array #12 | 6.19 | 17.64 | 408.93 |
| Parkfield | 1966 | San Luis Obispo | 6.19 | 63.34 | 493.5 |
| Parkfield | 1966 | Temblor pre-1969 | 6.19 | 15.96 | 527.92 |
| Borrego Mtn | 1968 | Pasadena—CIT Athenaeum | 6.63 | 207.14 | 415.13 |
| Borrego Mtn | 1968 | San Onofre—So Cal Edison | 6.63 | 129.11 | 442.88 |

Figure 3 shows the IS 1893:2016 target spectra associated with 5% damping and mean spectra of ground excitations. According to ASCE 7–16, the average spectrum must not fall below 90% of the target spectrum during the entire period range. From the figure, it can be observed that the mean spectra are well above 90% of the target spectra. Figures 4 and 5 show the first three linear mode shapes of five- and ten-story building models, respectively, in both directions. Tables 3 and 4 show the modal periods and the cumulative modal mass participation ratios of building models, respectively.

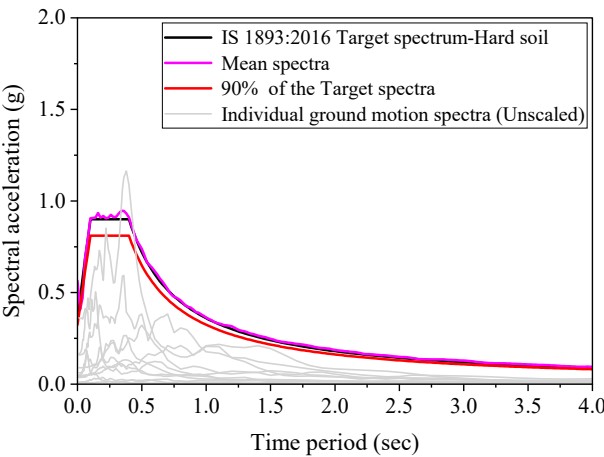

**Figure 3.** Target and mean acceleration spectra.

**Table 3.** Modal periods (seconds) of building models.

|  | 1(Y) | 2(X) | 3(R) | 4(Y) | 5(X) | 6(R) | 7(Y) | 8(X) | 9(R) | 10(Y) | 11(X) | 12(R) |
|---|---|---|---|---|---|---|---|---|---|---|---|---|
| M$_{5ref}$ | 0.54 | 0.44 | 0.42 | 0.18 | 0.14 | 0.13 | 0.1 | 0.08 | 0.07 | 0.06 | 0.05 | 0.11 |
| M$_{5bs5}$ | 0.76 | 0.58 | 0.56 | 0.23 | 0.18 | 0.17 | 0.12 | 0.1 | 0.09 | 0.08 | 0.06 | 0.05 |
| M$_{5ms5}$ | 0.71 | 0.56 | 0.53 | 0.2 | 0.16 | 0.15 | 0.14 | 0.1 | 0.09 | 0.08 | 0.07 | 0.06 |
| M$_{5ts5}$ | 0.59 | 0.47 | 0.45 | 0.25 | 0.19 | 0.18 | 0.14 | 0.11 | 0.1 | 0.09 | 0.07 | 0.06 |
| M$_{5bs7}$ | 1.11 | 0.81 | 0.76 | 0.25 | 0.2 | 0.19 | 0.13 | 0.1 | 0.09 | 0.08 | 0.07 | 0.06 |
| M$_{5ms7}$ | 0.96 | 0.73 | 0.68 | 0.22 | 0.17 | 0.16 | 0.15 | 0.12 | 0.11 | 0.08 | 0.07 | 0.06 |
| M$_{5ts7}$ | 0.67 | 0.53 | 0.5 | 0.34 | 0.25 | 0.24 | 0.14 | 0.11 | 0.1 | 0.08 | 0.06 | 0.05 |
| M$_{10ref}$ | 1.13 | 0.94 | 0.83 | 0.37 | 0.3 | 0.27 | 0.21 | 0.17 | 0.16 | 0.15 | 0.12 | 0.11 |
| M$_{10bs5}$ | 1.37 | 1.11 | 0.98 | 0.45 | 0.36 | 0.32 | 0.24 | 0.19 | 0.18 | 0.16 | 0.13 | 0.12 |
| M$_{10ms5}$ | 1.28 | 1.06 | 0.93 | 0.41 | 0.33 | 0.31 | 0.24 | 0.19 | 0.18 | 0.17 | 0.13 | 0.12 |
| M$_{10ts5}$ | 1.16 | 0.98 | 0.85 | 0.4 | 0.33 | 0.29 | 0.25 | 0.19 | 0.18 | 0.17 | 0.14 | 0.13 |
| M$_{10bs7}$ | 1.77 | 1.37 | 1.21 | 0.52 | 0.42 | 0.36 | 0.25 | 0.2 | 0.19 | 0.17 | 0.14 | 0.13 |
| M$_{10ms7}$ | 1.54 | 1.24 | 1.07 | 0.46 | 0.36 | 0.35 | 0.26 | 0.21 | 0.19 | 0.18 | 0.14 | 0.13 |
| M$_{10ts7}$ | 1.2 | 1.01 | 0.87 | 0.49 | 0.38 | 0.35 | 0.3 | 0.23 | 0.23 | 0.18 | 0.15 | 0.14 |

**Table 4.** Cumulative modal mass participation ratio of buildings.

|  |  | 1(Y) | 2(X) | 3(R) | 4(Y) | 5(X) | 6(R) | 7(Y) | 8(X) | 9(R) | 10(Y) | 11(X) | 12(R) |
|---|---|---|---|---|---|---|---|---|---|---|---|---|---|
| M$_{5ref}$ | UX | 0 | 83 | 83 | 83 | 94 | 94 | 94 | 94 | 97.8 | 97.8 | 97.8 | 97.8 |
|  | UY | 84.9 | 84.9 | 84.9 | 95.1 | 95.1 | 95.1 | 98.3 | 98.3 | 98.3 | 99.6 | 100 | 100 |
| M$_{5bs5}$ | UX | 0 | 94 | 94 | 94 | 99.2 | 99.2 | 99.2 | 99.2 | 99.8 | 99.8 | 99.8 | 99.8 |
|  | UY | 96 | 96 | 96 | 99.5 | 99.5 | 99.5 | 99.9 | 99.9 | 99.9 | 99.9 | 99.9 | 99.9 |
| M$_{5ms5}$ | UX | 0 | 76.3 | 76.3 | 76.3 | 94.7 | 94.7 | 94.7 | 97 | 97 | 97 | 97 | 97 |
|  | UY | 76 | 76 | 76 | 95.6 | 95.6 | 95.6 | 96.9 | 96.9 | 96.9 | 99.6 | 99.6 | 99.6 |
| M$_{5ts5}$ | UX | 0 | 78.7 | 78.7 | 78.7 | 90.4 | 90.4 | 90.4 | 96.6 | 96.6 | 96.6 | 96.6 | 96.6 |
|  | UY | 80.1 | 80.1 | 80.1 | 91.2 | 91.2 | 91.2 | 97.2 | 97.2 | 97.2 | 99.9 | 99.9 | 99.9 |
| M$_{5bs7}$ | UX | 0 | 98.1 | 98.1 | 98.1 | 99.8 | 99.8 | 99.8 | 99.9 | 99.9 | 99.9 | 99.9 | 99.9 |
|  | UY | 99 | 99 | 99 | 99.9 | 99.9 | 99.9 | 99.9 | 99.9 | 99.9 | 100 | 100 | 100 |
| M$_{5ms7}$ | UX | 0 | 69.7 | 69.7 | 69.7 | 69.7 | 95.5 | 95.5 | 96.4 | 96.4 | 96.4 | 96.4 | 96.4 |
|  | UY | 68.8 | 68.8 | 68.8 | 96.6 | 96.6 | 96.6 | 96.9 | 96.9 | 96.9 | 99.9 | 100 | 100 |
| M$_{5ts7}$ | UX | 0 | 78.7 | 78.7 | 78.7 | 90.4 | 90.4 | 90.4 | 96.6 | 96.6 | 96.6 | 96.6 | 96.6 |
|  | UY | 80.1 | 80.1 | 80.1 | 91.2 | 91.2 | 91.2 | 97.2 | 97.2 | 97.2 | 99.4 | 99.9 | 99.9 |
| M$_{10ref}$ | UX | 0 | 78.6 | 78.6 | 78.6 | 91.3 | 91.3 | 91.3 | 95 | 95 | 95 | 95 | 97 |
|  | UY | 80.7 | 80.7 | 80.7 | 92.3 | 92.3 | 92.3 | 95.8 | 95.8 | 95.8 | 97.5 | 98.6 | 98.6 |
| M$_{10bs5}$ | UX | 0 | 87 | 87 | 87 | 97.5 | 97.5 | 97.5 | 99.1 | 99.1 | 99.1 | 99.6 | 99.6 |
|  | UY | 90.3 | 90.3 | 90.3 | 98.9 | 98.9 | 98.9 | 100 | 100 | 100 | 100 | 100 | 100 |
| M$_{10ms5}$ | UX | 0 | 74.5 | 74.5 | 74.5 | 91.6 | 91.6 | 91.6 | 93.4 | 93.4 | 93.4 | 97.1 | 97.1 |
|  | UY | 75.5 | 75.5 | 75.5 | 92.6 | 92.6 | 92.6 | 93.9 | 93.9 | 93.9 | 97.6 | 97.6 | 97.6 |
| M$_{10ts5}$ | UX | 0 | 77.8 | 77.8 | 77.8 | 90.1 | 90.1 | 90.1 | 93.9 | 93.9 | 93.9 | 96.3 | 96.3 |
|  | UY | 79.8 | 79.8 | 79.8 | 90.9 | 90.9 | 90.9 | 94.4 | 94.4 | 94.4 | 96.8 | 96.8 | 96.8 |
| M$_{10bs7}$ | UX | 0 | 93.8 | 93.8 | 93.8 | 99.4 | 99.4 | 99.4 | 99.8 | 99.8 | 99.8 | 99.9 | 99.9 |
|  | UY | 96.4 | 96.4 | 96.4 | 99.6 | 99.6 | 99.6 | 99.8 | 99.8 | 99.8 | 99.9 | 99.9 | 99.9 |
| M$_{10ms7}$ | UX | 0 | 69.1 | 69.1 | 69.1 | 91.9 | 91.9 | 91.9 | 92.3 | 92.3 | 92.3 | 97.1 | 97.1 |
|  | UY | 68.5 | 68.5 | 68.5 | 92.7 | 92.7 | 92.7 | 92.9 | 92.9 | 92.9 | 97.5 | 97.5 | 97.5 |
| M$_{10ts7}$ | UX | 0 | 76.1 | 76.1 | 76.1 | 87.1 | 87.1 | 87.1 | 93 | 93 | 93 | 96 | 96 |
|  | UY | 77.7 | 77.7 | 77.7 | 87.2 | 87.2 | 87.2 | 93.7 | 93.7 | 93.7 | 96.6 | 96.6 | 96.6 |

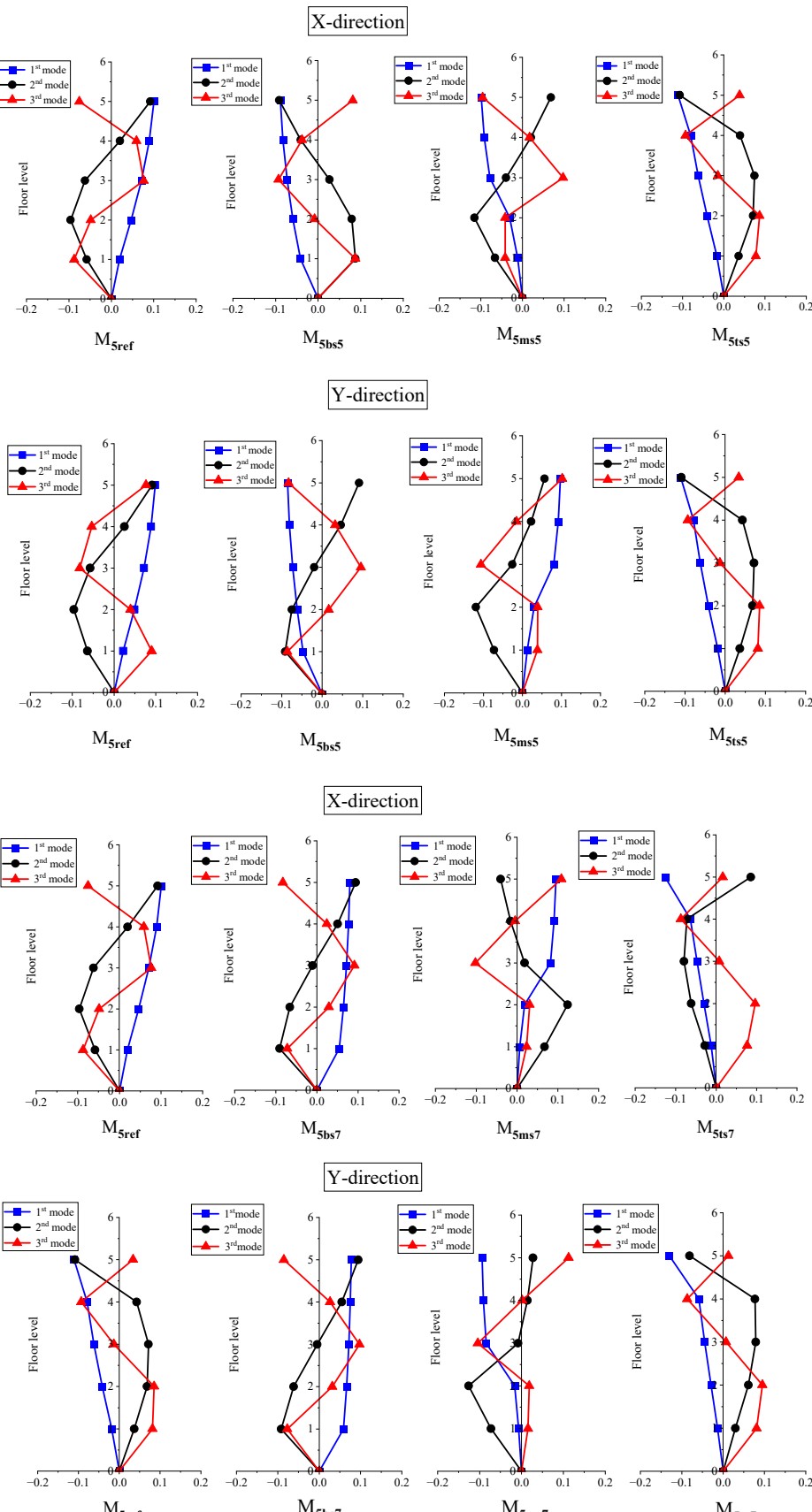

**Figure 4.** Linear mode shapes of five-story building models.

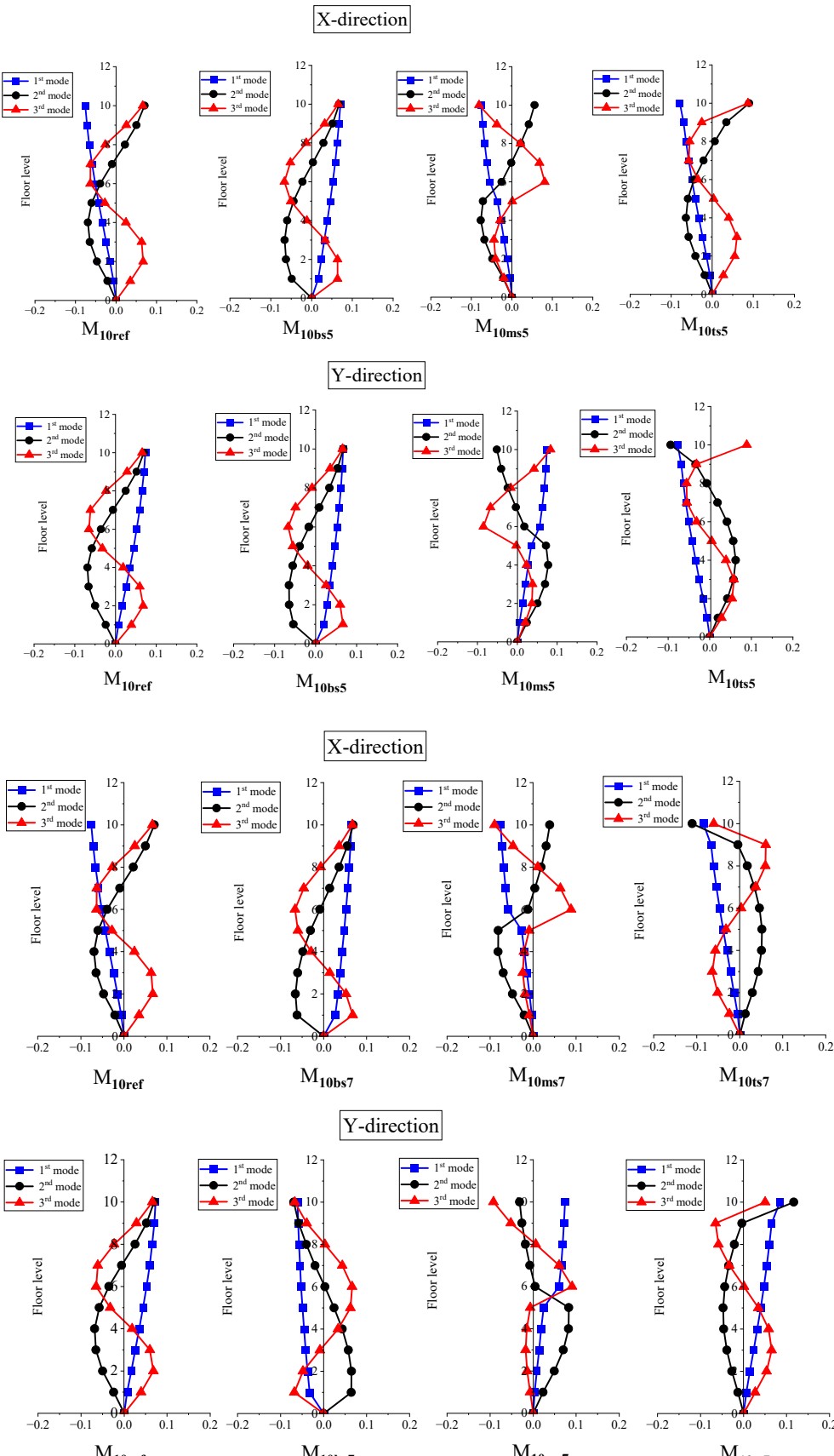

**Figure 5.** Linear mode shapes of ten-story building models.

## 4. Results and Discussion

The sections that follow investigate the behavior of the considered building models and the non-structural components. Displacements and inter-story drift ratios are the two response parameters that assess the structural behavior of the building models. Floor response spectra (FRS), floor amplification factors (FAF), peak component acceleration factors (PCAF), and component dynamic amplification factors (CDAF) are the response parameters that clearly describe the behavior of NSCs. Due to the application of bi-directional ground motion, all the response parameters are represented in two directions (X and Y) separately. For brevity, the response parameters are studied for 1st-, 3rd-, 5th-, 6th-, 8th-, and 10th-story levels in the case of 10-story building models.

### 4.1. Peak Story Displacement

Peak displacement patterns of the considered building models are presented in Figure 6. Primarily, it can be observed that the peak displacement values increase with the height of the building models. The peak displacements of a reference building are lower than that of the buildings with a soft story at different levels. The measured peak displacements suddenly increase at that particular floor level where a soft story exists, as can be seen from Figure 6. The building model with a soft story at the bottom level shows a more significant displacement at all floors compared to that of the remaining building models. In the case of a top soft-story building, the behavior of the building is equivalent to the reference building at all story levels except at the top story. In the case of a building with a soft story at the middle floor level (i.e., at the third floor in the case of a five-story building, and sixth floor in the case of a ten-story building), the increase in story displacements compared to that of the reference building is observed from the soft-story level and above. A similar research outcome was obtained by Alghany et al. in their most recent study [39]. As the soft-story height increases from 5 m to 7 m, the peak displacements increase for all the building models.

The existence of a soft story at different levels has an insignificant effect on the top-story displacement of building models. The decrease in the top-floor displacement with a soft-story height of 5 m is 20.5% and 21.9% in X and Y directions, respectively, compared to the soft-story height of 7 m in the bottom soft-story building model (five-story). Similarly, the decrease in top-floor displacement with a soft-story height of 5 m is 9.84% and 13.58% in X and Y directions, respectively, compared to a soft-story height of 7 m in the bottom soft-story building model (ten-story). From Figure 6, it can be observed that the displacement of a building model with a soft story at the bottom level is approximately three and seven times higher than that of the reference building at the respective floor level in the X direction for a soft-story height of 5 m and 7 m, respectively. The increase in the displacements of a building model with a soft story at the middle- and top-floor levels is smaller than that of the building with a soft story at the bottom floor level with respect to the reference building at the respective floor level. Thus, it can be concluded from the above observations that the building model with a bottom soft story exhibits a considerable vertical stiffness irregularity [32] and follows the middle soft-story building model.

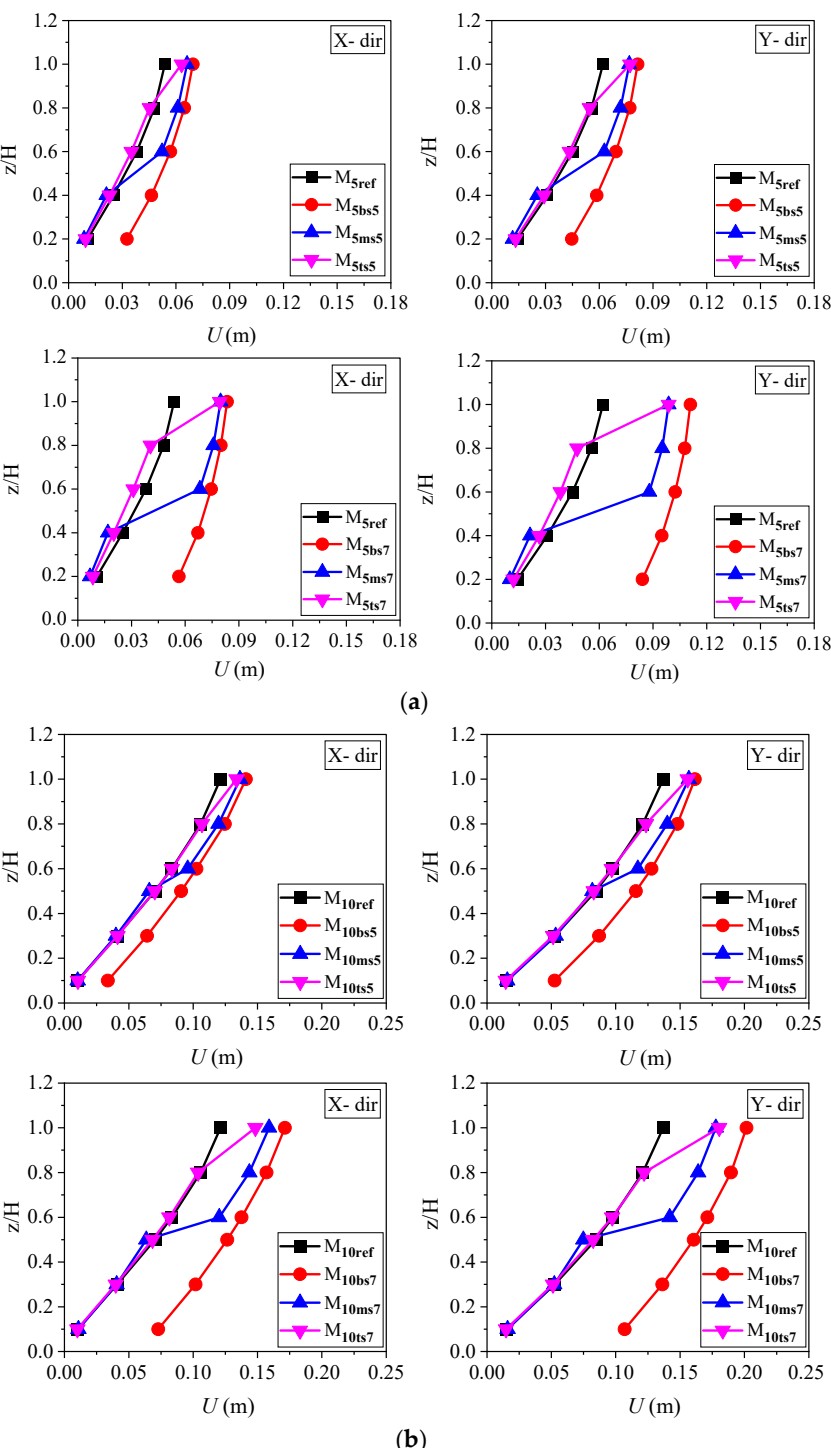

**Figure 6.** Displacement of a building models: (**a**) five-story; (**b**) ten-story.

### 4.2. Inter-Story Drift Ratio (IDR)

Drift ratio (Δ) in this study refers to the difference between a story's displacement with respect to the immediately lower one divided by the distance between that story and the lower one.

Almost all structural elements in a structure are subject to inter-story drift ratio (IDR), making it one of the crucial engineering demand parameters (EDPs). Building occupant safety may be compromised by damage to any structural component during an earthquake. As a result, this study examines the impact of the position of soft stories in structures on IDR peaks. Inter-story drift ratios of the considered building models are presented in

Figure 7. It has been noted that the magnitude of the peak IDR is more significant in the soft-story building models at the soft-story level than that of the reference building model in both low- and high-rise structures (Figure 7a,b, respectively). It can also be deduced from the figure that the IDR values in the Y direction are larger than those in the X direction in all the considered models.

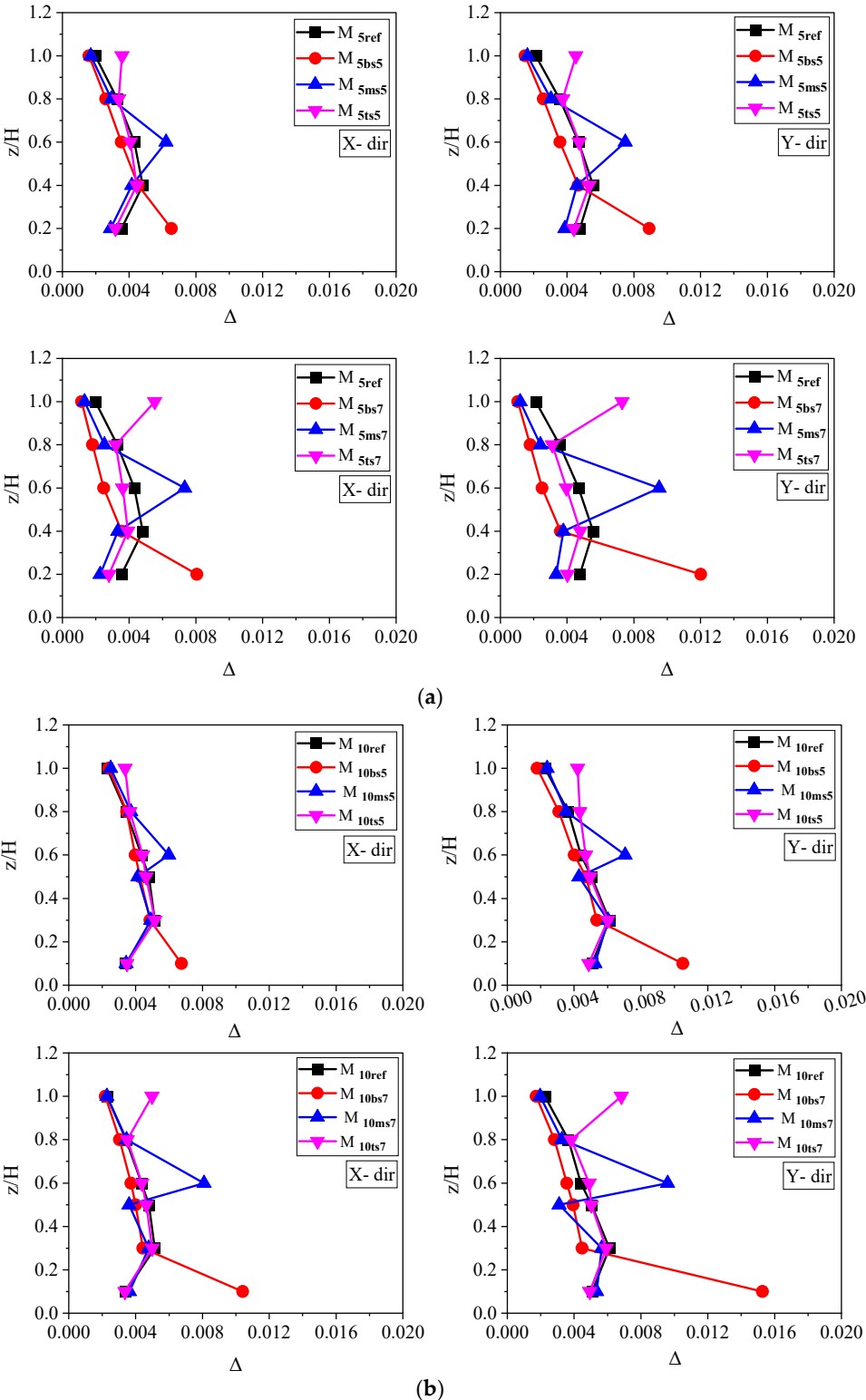

**Figure 7.** Inter-story drifts of building model: (**a**) five-story; (**b**) ten-story.

The IDR values increase as soft-story height increases from 5 m to 7 m. All of the models exhibit a similar pattern of peak IDR variation along the relative height of the building. The maximum inter-story drifts for a five-story building model from Figure 7a are 44.5 mm and 60 mm when the soft story is located at the bottom floor in the Y direction for the height of the soft story of 5 m and 7 m, respectively. Similarly, the corresponding drift values for a ten-story building model (Figure 7b) are 52.5 mm and 106.89 mm for the height of the soft story of 5 m and 7 m, respectively. Thus, it can be concluded from the analysis results that, compared to the reference building model, the building models with a soft story have a sudden increase in the IDR at the specified story level in two orthogonal directions. The maximum rapid change in drift values is observed in a building with a soft story at the bottom floor level compared to the other models.

### 4.3. Floor Response Spectra (FRS)

The NSCs studied in the present research are elastic single degree-of-freedom (SDOF) systems. The mass of the NSC is assumed to be small compared to that of the main structure (dynamic interaction is neglected). The use of FRS is a decoupled method that individually assesses the structure and NSC in a specified pattern. Scaled ground motions are used as the input for the linear time history analysis. Absolute acceleration responses are obtained from the models at all the floors individually and used as an input for NSC to generate corresponding FRS. In the abovementioned specified pattern, floor response spectra were obtained for all the considered models. Floor acceleration time histories in X and Y directions are utilized to get FRS. These FRS were obtained at a 5% damping ratio, and the mean results are plotted for each floor.

The mean spectral acceleration ($S_a$ in $g$ units) of an NSC attached to the floor is plotted against the vibration period ($T_s$ in seconds) for all the considered five- and ten-story building models shown in Figures 8 and 9, respectively, in both X and Y orthogonal directions. Maximum spectrum peaks should arise at the fundamental natural period of the supporting structure when the FRS are plotted across a wide range of periods [58]. The peaks observed in the FRS are consistent with the modal periods of the corresponding building models. It is noticed that the spectral accelerations in the X direction are high compared to those in the Y direction in all the considered models corresponding to the first modal period. The magnitude of FRS at all floor levels of the building models decreases as the soft-story height increases from 5 m to 7 m. The magnitude of FRS increases from the lower first floor to the upper fifth floor in all the models. It is worth mentioning that the presence of a soft story and its position in the building model has a significant effect on the FRS and can be observed from the FRS curves presented in Figures 8 and 9. In a five-story reference building, two peaks in the FRS can be observed and are consistent with the first and second modal periods of a building for the lower floor levels (first and second floors), and the contribution of higher mode is insignificant for the top floor levels (third, fourth and fifth floors). This observation is consistent with one of the research outcomes ascertained by Berto et al. [19]. As the building height increases (ten-story reference building), the peak spectral acceleration associated with the higher modes increases. The second and third modes have a significant effect on the peaks of FRS until the third-floor level, and the contribution of the higher mode (third mode) shows an insignificant impact on the FRS as the floor level increases. Thus, it can be concluded that in a building without any vertical stiffness irregularity, the short-period NSCs will experience a high seismic demand when attached to the lower floor levels.

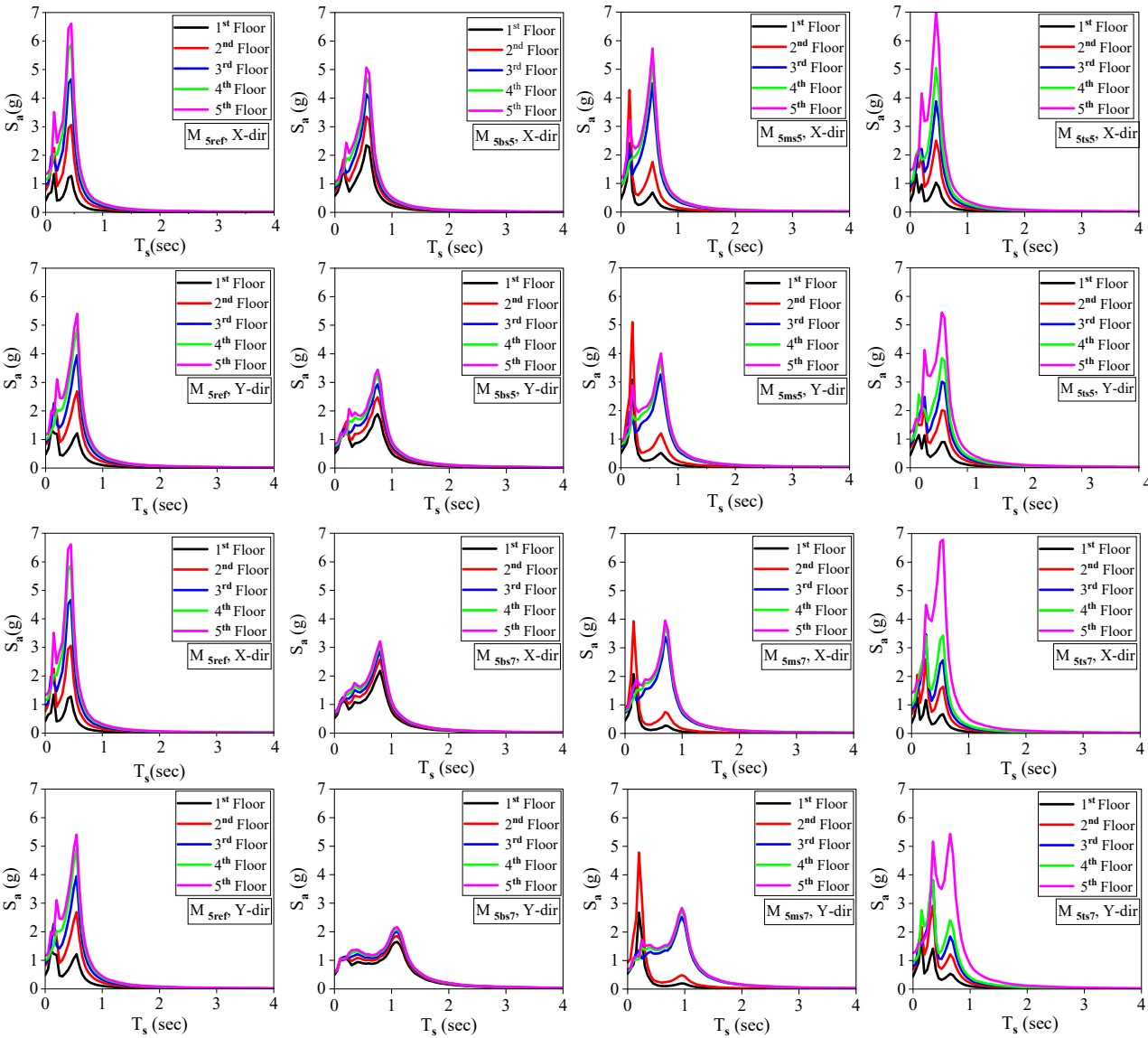

**Figure 8.** Floor response spectra of the five-story building models.

In the bottom soft-story building, the FRS of a specified floor is higher than that of the reference building in both X and Y orthogonal directions. The amplification of FRS peaks associated with modal periods follows a similar trend to that observed in the reference building. Building models exhibit more substantial floor-to-floor amplification of spectral accelerations when the soft-story height is 5 m. As the soft-story height increases to 7 m, the spectra on various levels merge. The peak spectral acceleration of the bottom floor (first floor) associated with the first modal period in the model $M_{5bs5}$ is increased by 79% and 55.3% as compared to the $M_{5ref}$ in X and Y directions, respectively. Similarly, the peak spectral acceleration in the X and Y directions, respectively associated with the second modal period in the $M_{10bs5}$, is increased by 67.9% and 48.6% as compared to the $M_{10ref}$. In X and Y directions, the spectral acceleration of the model $M_{5bs7}$ increased by 71.6% and 33%, when compared to $M_{5ref}$. In the case of a top soft-story building, the peak spectral acceleration of the top floor associated with the first modal period in the model $M_{5ts5}$ is increased by 6.36% and 0.55% as compared to the $M_{5ref}$ in X and Y directions, respectively. Similarly, the peak spectral acceleration in the X and Y directions, respectively associated with the second modal period in the $M_{10ts5}$, is increased by 23.28% and 49.8% as compared to the $M_{10ref}$. In X and Y directions, the spectral acceleration of the model $M_{5ts7}$ increased by 2.72% and 0.37%, respectively, when compared to $M_{5ref}$.

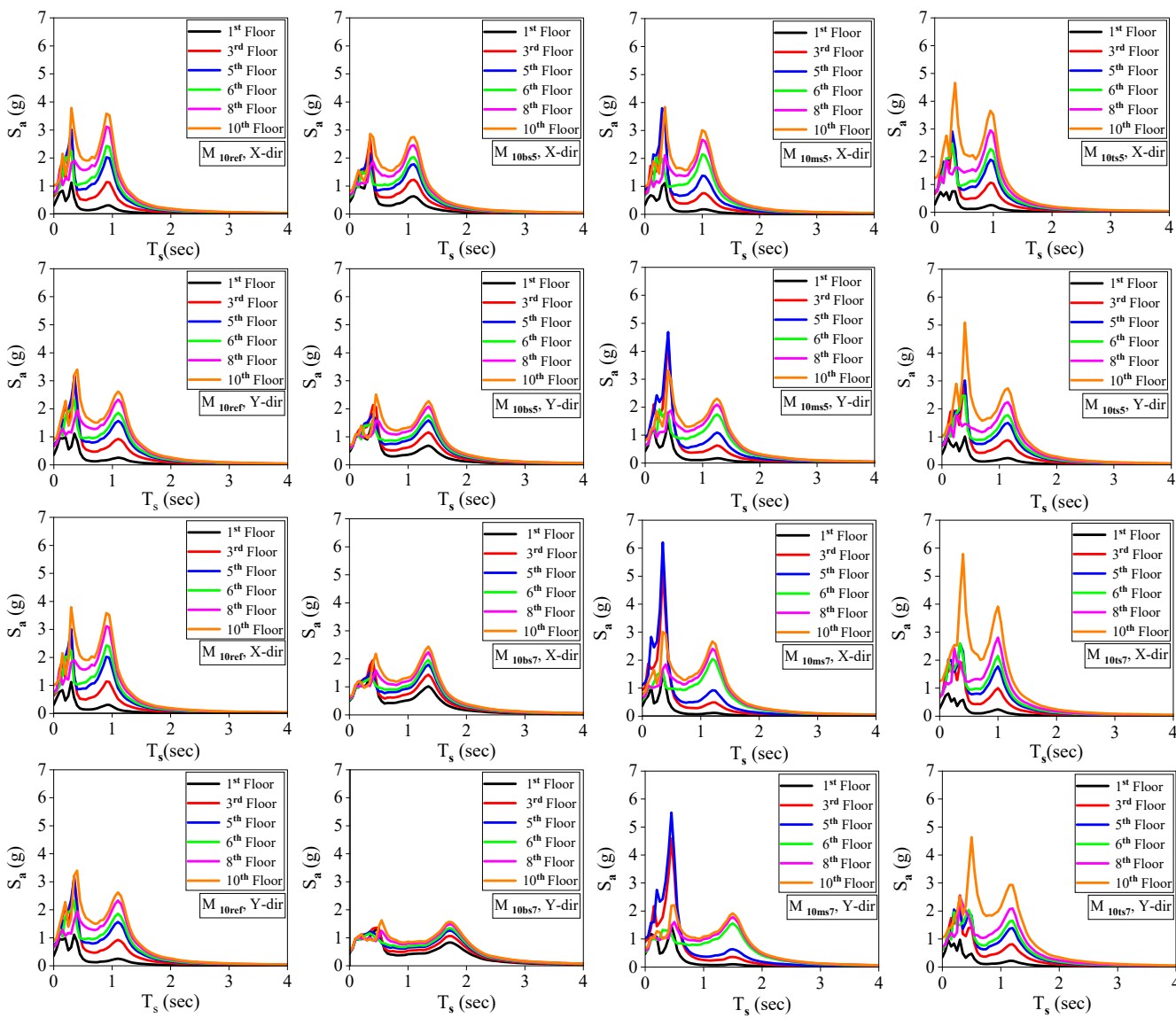

**Figure 9.** Floor response spectra of the ten-story building models.

In the middle soft-story building model, the presence of a soft story shows a significant effect on the peak spectral acceleration of the immediate floor level below the soft-story level, i.e., (second and fifth-floor levels in five- and ten-story building models, respectively). The peak spectral acceleration associated with the second modal period is maximum in all the building models with different soft-story heights. For instance, in the X direction, the peak spectral acceleration of the second floor associated with the first modal period of the model $M_{5ms5}$ is reduced by 42.8%, and that associated with the second modal period is raised by 89.7% when compared to the $M_{5ref}$. Similarly, in the case of the ten-story building model, the peak spectral acceleration of the fifth floor associated with the second modal period of the model $M_{10ms5}$ is raised by 21.6% compared to the $M_{10ref}$. Thus, in the situation of a middle soft-story building, a component whose vibration period is expected to correspond with the first modal period of vibration of the building model might be attached to floors below the soft-story level where the first mode of vibration has little effect.

### 4.4. Evaluation of Floor Amplification

The current study assesses the amplification in floor acceleration. For this purpose, peak floor acceleration (PFA) normalized with peak ground acceleration (PGA) is defined and plotted against the height of a building. Figure 10 shows the variation of normalized floor acceleration with the relative height of a building.

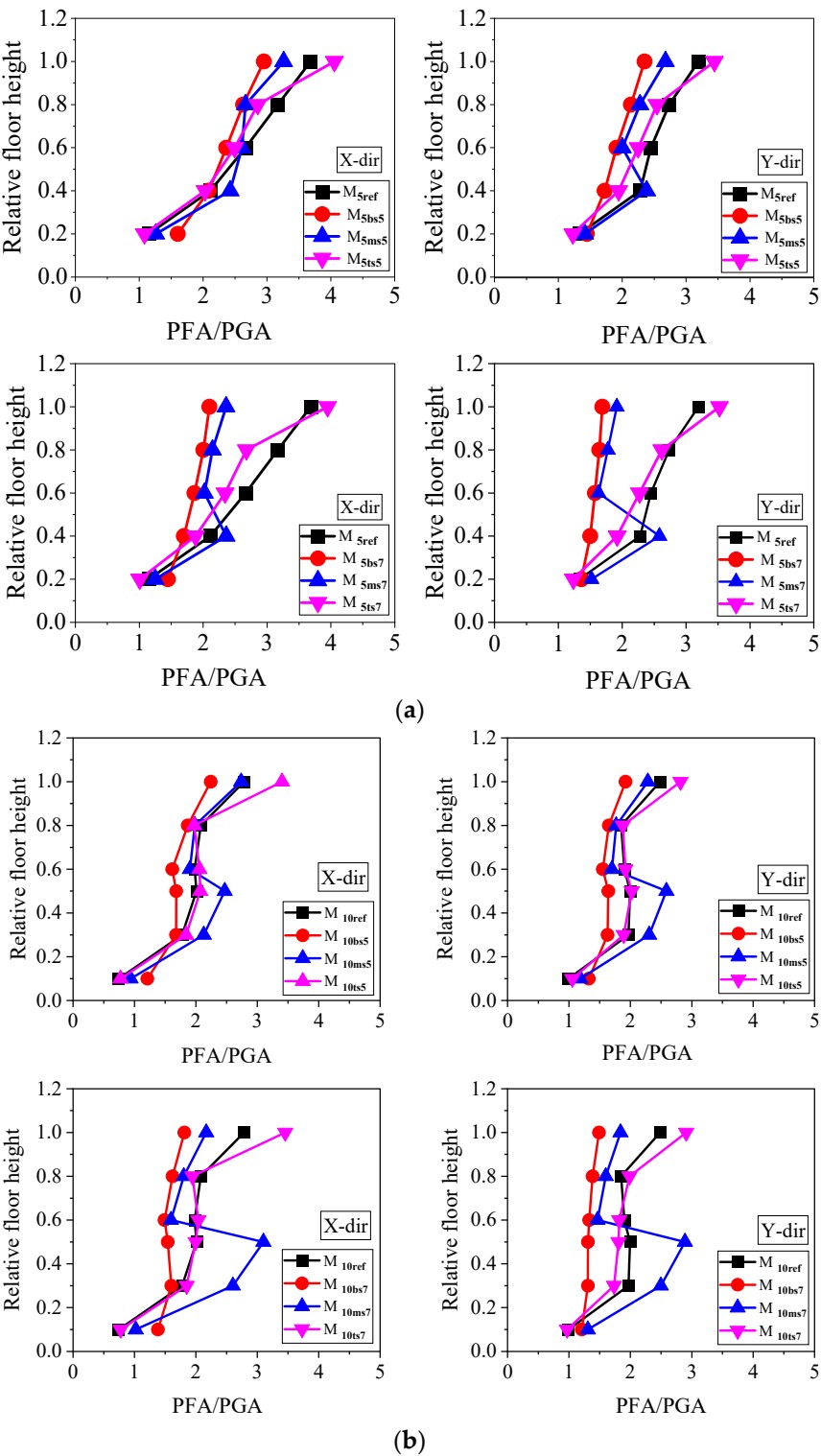

**Figure 10.** Variation of floor amplification factors with the building height for (**a**) five-story and (**b**) ten-story building models.

The amplification of floor acceleration in the soft-story building models is more than the reference building models at the soft-story level for bottom and top soft-story building models in both orthogonal directions. From Figure 10, it can be observed that the values of the ratio $PFA/PGA$ of the bottom floor of the model $M_{5bs}$ increased by 39.7% and 26.1% for soft-story heights of 5 m and 7 m, respectively, in the X direction when compared to the model $M_{5ref}$. The increase of 9.1% and 7.2% in the ratio $PFA/PGA$ is observed for soft-story heights of 5 m and 7 m, respectively, in the top floor of $M_{5ts}$ in the X direction compared with the $M_{5ref}$. In the case of $M_{10bs}$, the values of the ratio $PFA/PGA$ in the bottom increased by 62.6% and 85.5% for soft-story heights of 5 m and 7 m, respectively, in the X direction when compared to the model $M_{10ref}$. The increase of 22.2% and 24.1% in the ratio $PFA/PGA$ is observed for soft-story heights of 5 m and 7 m, respectively, in the top floor of $M_{10ts}$ in the X direction compared with the $M_{10ref}$.

In the middle soft-story building model, the floors below the soft-story level exhibit more amplification in floor acceleration than the reference building models. For instance, in the five-story, middle soft-story building, the values of ratio $PFA/PGA$ in the second floor increased by 14% and 11.8% for soft-story heights of 5 m and 7 m, respectively, when compared with the reference building in the X direction. Similarly, the values of the ratio $PFA/PGA$ in the fifth floor increased by 22.8% and 54.2% for soft-story heights of 5 m and 7 m, respectively, when compared with the reference building in the X direction in the case of the ten-story building model. As a consequence of the analysis results in this section, it can be inferred that the location of a soft story has a considerable influence on the peak floor accelerations.

Several code formulas exist to assess the variation of peak floor acceleration along with the structure's height. The floor amplification factor (PFR/PGA) specifications for several seismic codes, including ASCE 7–16 [29] and Eurocode 8 [59], are provided by given Equations (1) and (2), respectively.

$$PFA/PGA = 1 + 2\frac{z}{h} \tag{1}$$

$$PFA/PGA = 1 + 1.5\frac{z}{h} \tag{2}$$

From Figure 11, it can be observed that the code formulations show a linear variation of floor amplification with the height of a structure. However, the analysis found that the variation of floor acceleration throughout the building height is non-linear. It is also worth pointing out that the code formulations underestimate the PFA demands at a soft-story level in all the considered building models. Therefore, it can be inferred that the linear hypothesis in the code-based formulae may result in an under- or overestimation of PFA demands. The current code-based formulae may be modified by incorporating the effects of vertical stiffness irregularity into the analysis, as their impact can be seen on the PFA demands. In light of this, it can be said that the code-based formulations do not adequately estimate the peak response of non-structural components together with building height.

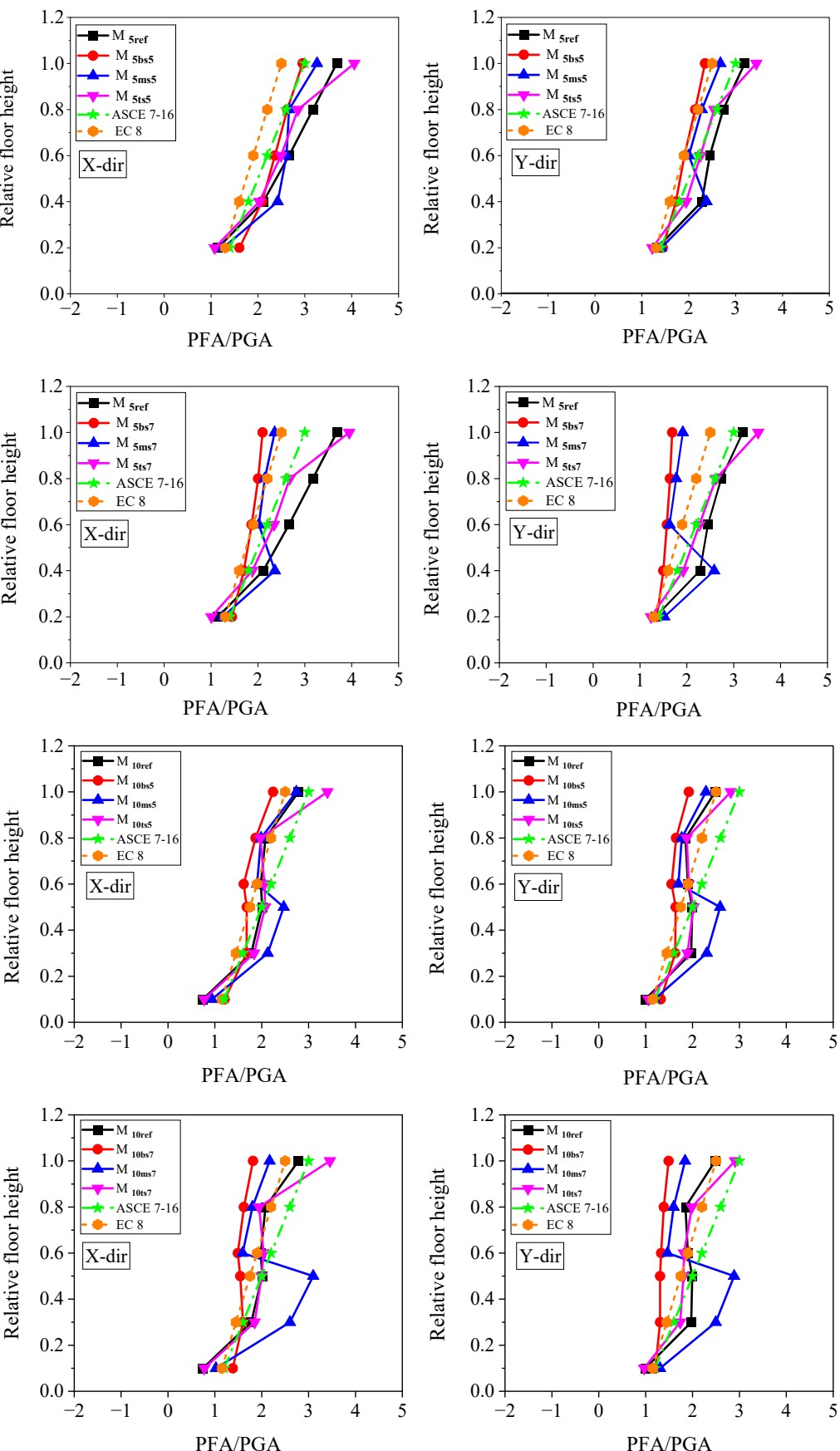

**Figure 11.** Comparison of floor amplification factors between the considered building models and current codes.

*4.5. Peak Component Acceleration*

The maximum ordinate in the floor response spectrum is called peak component acceleration (PCA). In the current study, the PCA is normalized with the PGA, and the ratio $PCA/PGA$ is plotted against the relative height of a building model.

Figure 12 shows the variation of normalized PCA with the height of the building models. From Figure 12, it can be observed that the behavior of the components is different in the two orthogonal directions. The peak acceleration of the component is higher in the vertical stiffness of irregular buildings when compared to the reference building at a soft-story level in the case of bottom and top soft-story models, and at floors below the soft-story level in the case of middle soft-story model. The linear variation of the ratio $PCA/PGA$ with the building height is observed for models $M_{5ref}$, $M_{5bs5}$, $M_{5ts5}$, and $M_{5bs7}$. Such linear variation is due to the fact that the effect of higher modes is insignificant. The first mode is the dominant one on the acceleration of the components, as observed in Figure 8. However, this is not the case with the models $M_{5ms5}$, $M_{5ms7}$, and $M_{5ts7}$, where the effect of higher modes is significant with respect to the behavior of the NSCs (Figure 8). In the case of ten-story building models, the variation of the ratio $PCA/PGA$ with the building's height is non-linear as expected since the participation of higher modes is significant in the tall buildings. The effect of a top soft story (height = 5 m) on the peak acceleration of the component is very minimal at the soft-story level in the case of the five-story building model.

As the soft-story height increases from 5 m to 7 m, the PCA values at the soft-story level (bottom and top soft-story models) and consecutive floors below the soft-story level (middle soft-story model) decrease in five-story building models. The increase in the soft-story height resulted in a reduction of the PCA value at the soft-story level of a model $M_{10bs}$. The increase in the magnitude of PCA values was observed in the model $M_{10ts}$ as the soft-story height increased from 5 m to 7 m. The consecutive floors below the soft-story level show an increasing trend in the values of PCA as the soft story rises from 5 m to 7 m in the $M_{10ms}$; therefore, NSCs attached to these floors are unsafe, and proper care must be taken. As a result of the findings in this section, it is reasonable to conclude that the position of a soft story and its height has a substantial influence on the component's peak acceleration.

From Figure 13, it can be observed that the formulation given by ASCE 7–16, as defined in Equation (3), underestimates the PCA demands along the building height in all the considered building models. The current code-based formulation should be modified by incorporating the effects of vertical stiffness irregularity into the analysis. Hence, it can be concluded that the code-based linear formulation cannot accurately estimate the peak acceleration response of the NSCs.

$$PCA/PGA = a_p\left(1 + 2\frac{z}{h}\right) \tag{3}$$

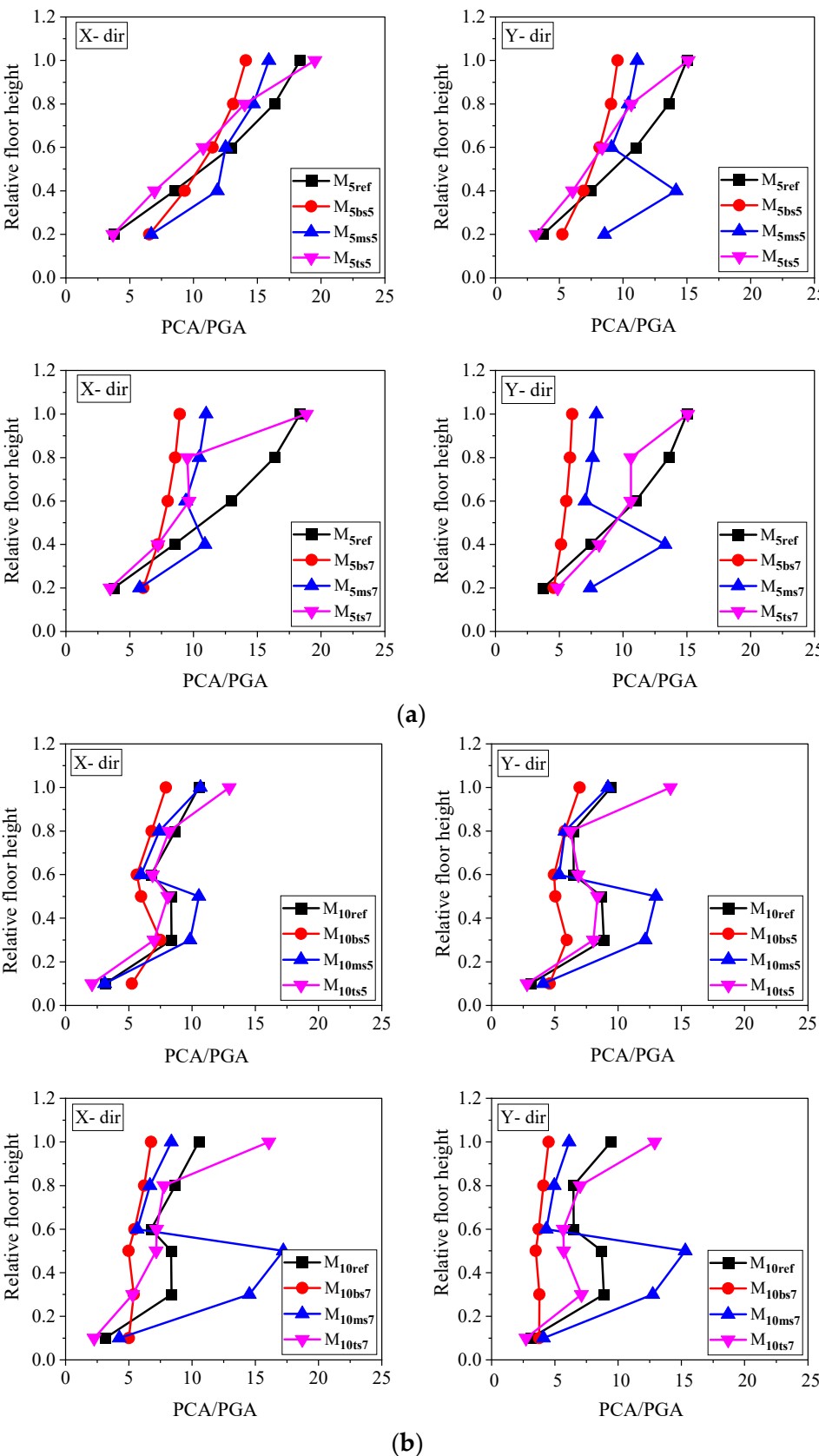

**Figure 12.** Variation of peak component amplification factors with the building height for (**a**) five-story and (**b**) ten-story building models.

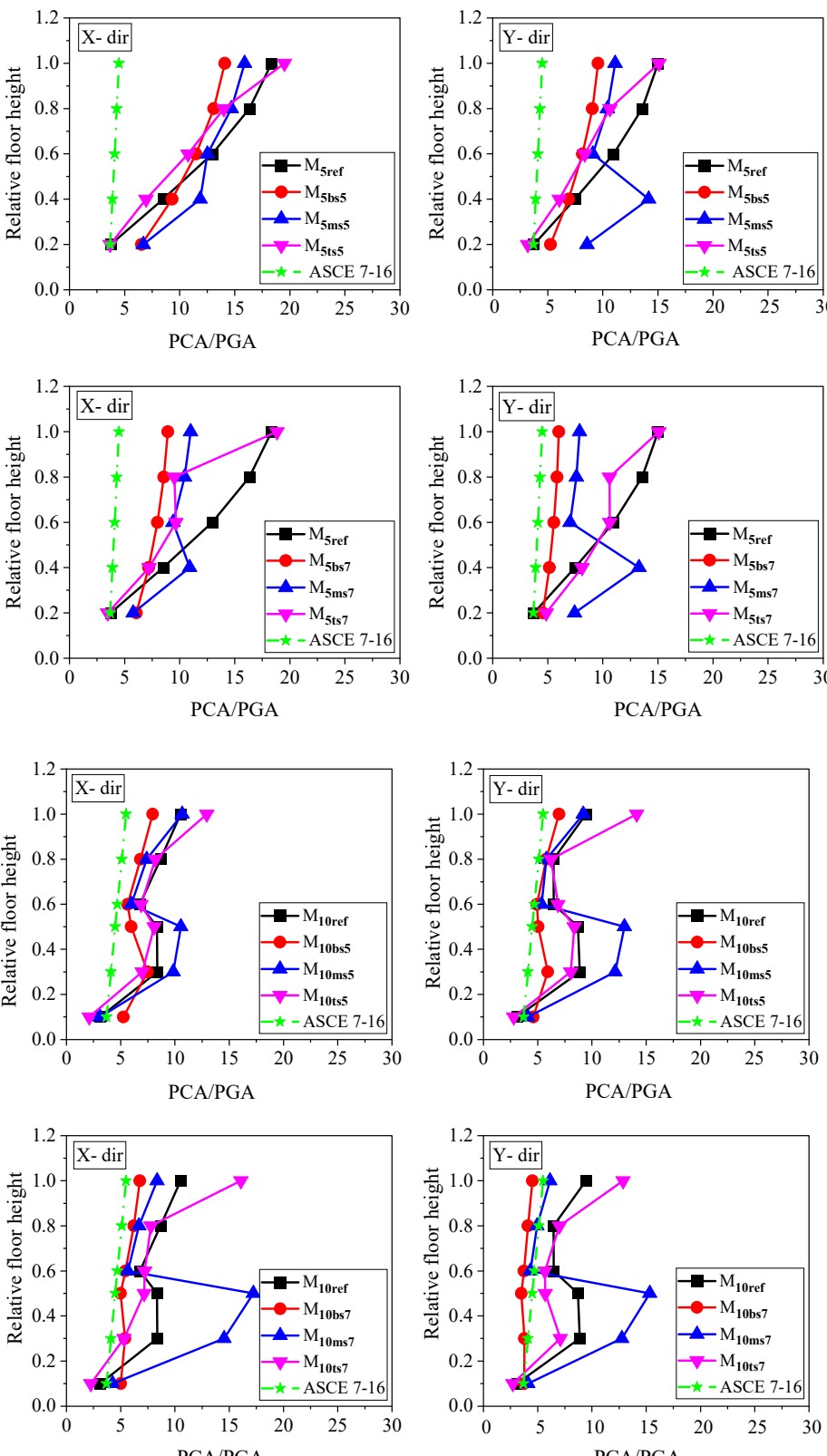

**Figure 13.** Comparison of normalized PCA values between the considered building models and current code.

### 4.6. Component Dynamic Amplification Factor

This section investigates the amplification in the acceleration of the component relative to the floor acceleration to which it is attached. The story FRSs of all the considered building models normalized by the corresponding PFAs are carried out. The FRS of the building models at a soft-story floor level normalized by the corresponding peak floor accelerations (PFAs) are shown in Figure 14. The ratio $FRS/PFA$ represents the component dynamic amplification factor (CDAF). The CDAF of the building models in the present study is compared with the definitions of ASCE 7–16 [29] and FEMA P-750 [60]. As per the definition of ASCE 7–16, the component amplification factor ($a_p$) is 2.5 for flexible NSCs whose time period is larger than 0.06 s. For rigid NSCs ($T < 0.06$ sec), the value of the amplification factor is 1.

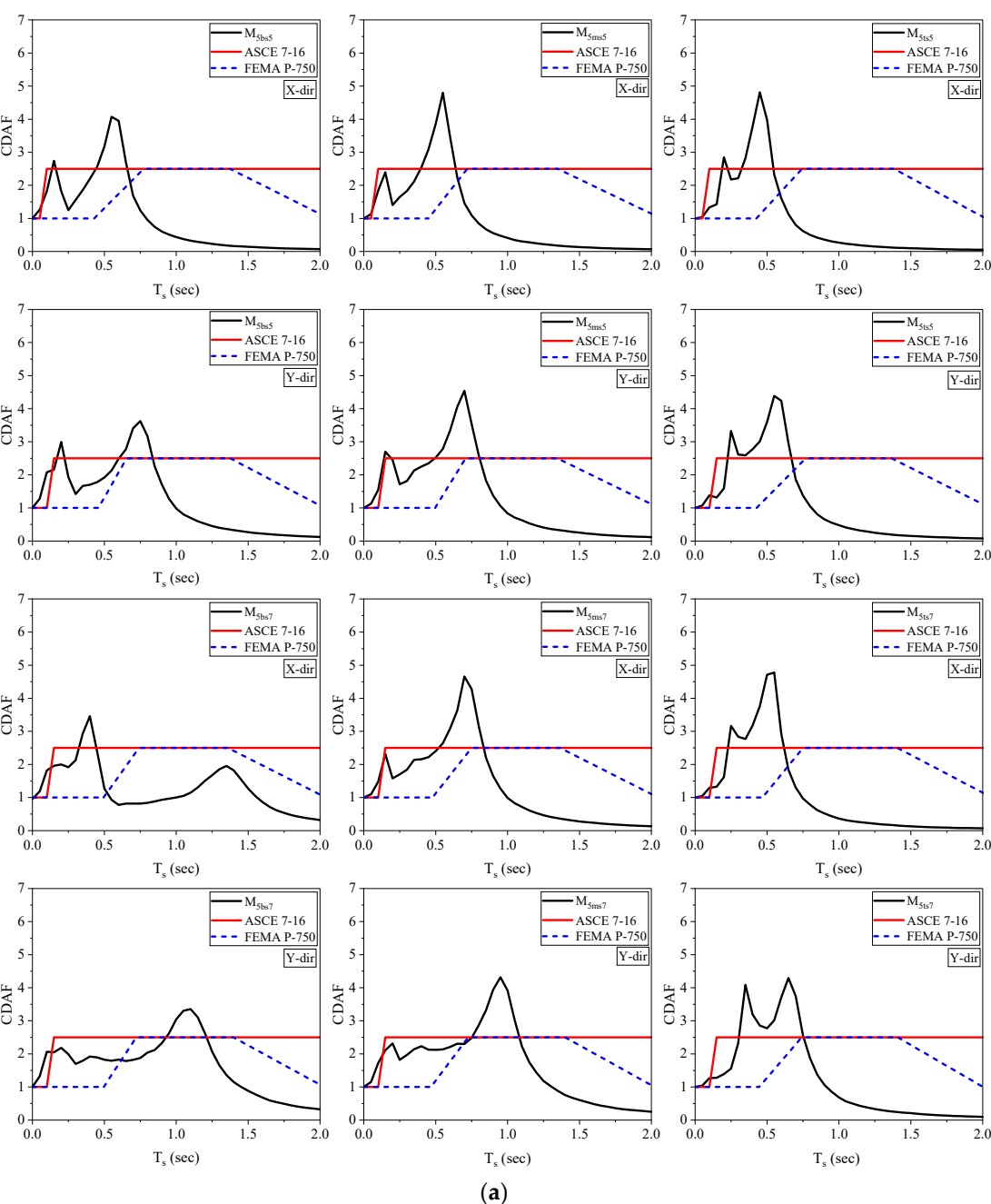

(**a**)

**Figure 14.** *Cont.*

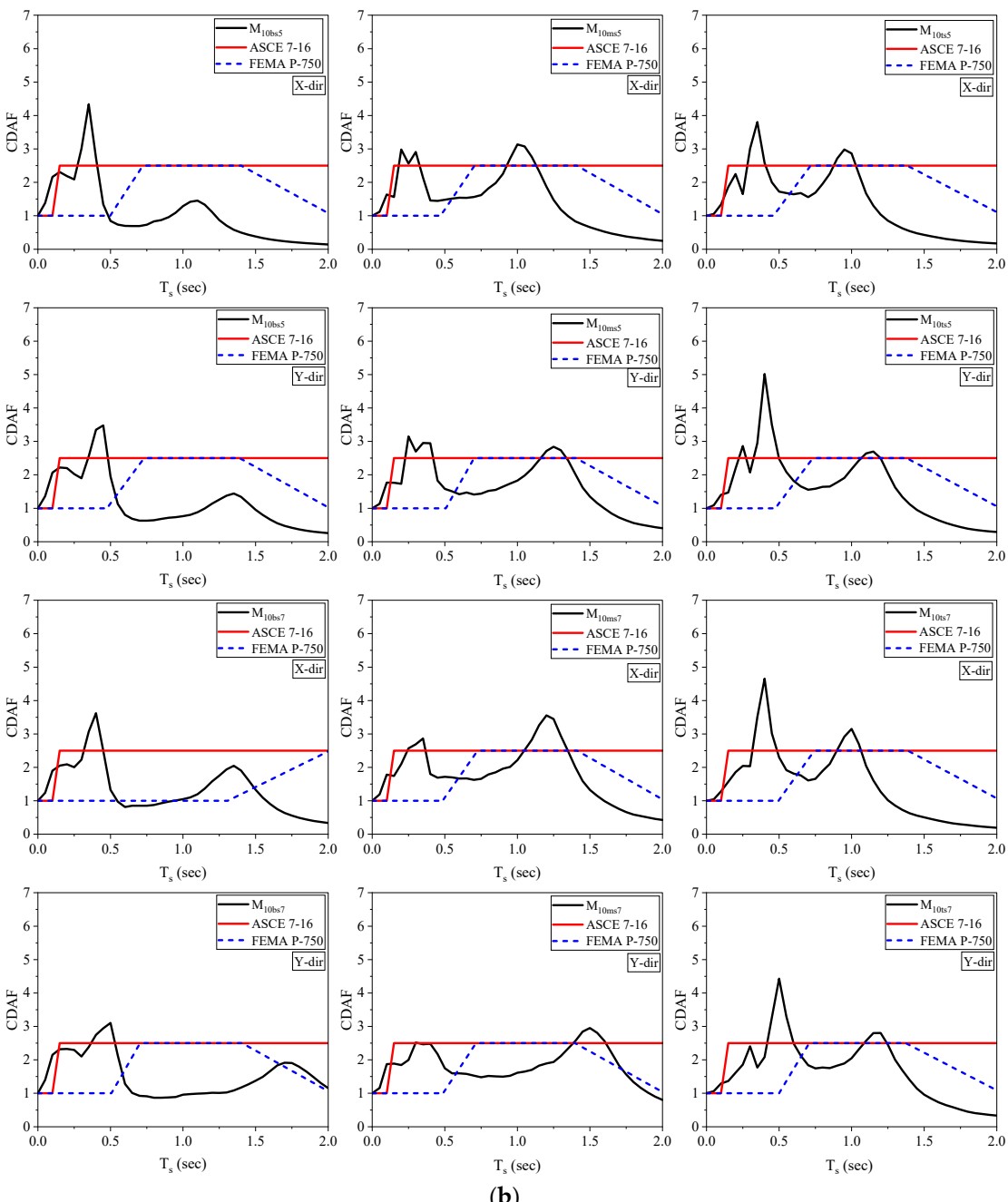

**Figure 14.** Component dynamic amplification factor values for (**a**) five-story building models and (**b**) ten-story building models.

The maximum value of the CDAF spectrum is termed the amplification factor. The amplification factor values at the soft-story level range from 3.55 to 4.92 for the middle soft-story building models, corresponding to the fundamental vibration period of the NSC. For the bottom soft-story building models at the same period, the values of the amplification factors vary between 1.37 and 4.07. The amplification factor values at the soft-story level for a fundamental vibration period of the NSC vary between 2.69 and 4.79 for top soft-story building models. As a result, it can be stated that the middle soft-story building models exhibit the greatest amplification in component acceleration.

From Figure 14, it is clear that the definitions of FEMA P-750 and ASCE 7–16 substantially underestimate the dynamic amplification factor in the building. The major explanation for this observation is that code models for estimating dynamic amplification factor are based on the assumption that the building response is dominated by the funda-

mental mode of vibration and that it varies linearly throughout the height of the building. In the zone of the fundamental mode of the building, the ASCE 7–16 and FEMA P-750 models are non-conservative. This non-conservatism is especially noticeable in the case of the five-story elastic supporting structure. In the case of the high-rise structure (ten-story), the ASCE 7–16 and FEMA P-750 models are non-conservative in the impact zone of the second mode of the building models. This result is consistent with earlier research on regular multi-story structures [13,61]. As a result, the present code-based formulation should be modified to account for the effects of vertical stiffness irregularity and higher modes in the analysis.

## 5. Summary and Conclusions

Non-structural components (NSCs) have become critical in sustaining post-earthquake functionality while constructing seismic-resilient structures. The present study was dedicated to assessing the effect of a soft story and its location on the seismic demands of a building structure and NSCs. The building structures considered are five-story and ten-story reinforced-concrete framed structures. The vertical stiffness irregularity (soft story) was considered at the bottom, middle, and top-story levels. Story displacements and inter-story drift ratios are evaluated to assess structural behavior. The floor response spectra and the amplification effects of NSC on the floor acceleration responses are studied to understand the behavior of NSCs. For the time history analysis, 11 ground motions are considered, making them spectrally compatible with the IS code-based design spectrum consistent with the hard soil and seismic zone V. Based on the analysis of the building models, the following conclusions can be drawn:

1. The existence of a soft story at different levels has an insignificant effect on the top-story displacement of the building models. The amplification in the peak story displacements of a building with a middle and top soft story was smaller than that of the building with a bottom soft story at the respective floor level. As the soft-story height increases, the peak story displacements increase for all the building models.

2. The magnitude of the peak inter-story drift ratio is more significant in the soft-story building models at the soft-story level than that of the reference building models. The building models with a soft story have a sudden increase in the drift ratio at the specified story level in the two orthogonal directions. The maximum rapid change in drift values was observed in a building with a bottom soft story compared to the other models.

3. The short-period NSCs will experience a high seismic demand when attached to the lower floor levels in a building without any vertical stiffness irregularity. Building models exhibit more substantial floor-to-floor amplification of spectral accelerations when the soft-story height is 5 m. As the soft-story height increases to 7 m, the spectra on various levels merge.

4. The floor amplification factors and normalized peak component accelerations were amplified at the soft-story level in the bottom and top soft-story models. In the case of middle soft-story buildings, such amplification was observed in the immediate floors below the soft-story level.

5. The middle soft-story buildings exhibit the greatest amplification in the component's acceleration (CDAF). The average amplification factor values at the soft-story level are 2.72, 3.74, and 4.24 for the bottom, top, and middle soft-story building models correspond to the fundamental vibration period of the NSC when considering all the building models and soft-story heights.

6. Code formulations underestimate the PFA demands at a soft-story level in all the considered building models. Therefore, it can be inferred that the linear hypothesis in the code-based formulae may result in an under- or overestimation of PFA demands together with building height.

7. The code definitions underestimate the peak component acceleration and dynamic amplification factors at the soft-story level. Hence, the current code-based formulation

should be modified by incorporating the effects of vertical stiffness irregularity into the analysis.

The observations made during this research are confined to the examined buildings and the ground motions. This study is limited to linear analysis as a preliminary investigation method. The nonlinear behavior of the building model needs to be considered for more generalized results. Future research can be extended to study high-rise structures such as 15, 20, or 30 stories with different irregularities. It is also worth mentioning that peak floor accelerations were computed and used to generate floor response spectra, which may be used to quantify seismic losses related to NSEs. The FEMA P-58 approach will be used in future studies for NSC loss calculation by using floor spectral acceleration as the reference engineering demand parameter (EDP).

**Author Contributions:** Conceptualization, S.P.C., V.P., I.H.; methodology, S.P.C., V.P., F.V.; software, S.P.C., V.P., F.V., M.J.; formal analysis, S.P.C., V.P., I.H., P.S.C.B.; writing—original draft preparation, S.P.C., V.P.; writing—review and editing, S.P.C., U.R., M.J., P.S.C.B.; supervision, S.P.C.; project administration, S.P.C. All authors have read and agreed to the published version of the manuscript.

**Funding:** This research received no external funding.

**Institutional Review Board Statement:** Not Applicable.

**Informed Consent Statement:** Not Applicable.

**Data Availability Statement:** Data available within the article.

**Conflicts of Interest:** The authors declare no conflict of interest.

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
