# Peer review of "Influence of a Soft Story on the Seismic Response of Non-Structural Components"

_sustainability, doi:10.3390/su15042860_

Round 1

Reviewer 1 Report

The paper presents a study on seismic floor demand in the case of soft-storey buildings. In my opinion, the paper should be improved according to the comments in the attached PDF file.

Reviewer 2 Report

The paper “Influence of Soft-Story on Seismic Response of Non-Structural Components” presents an analytical study investigating the effects of soft story on the seismic behavior of reinforced concrete (RC) buildings and non-structural components (NSCs). For this, 3D linear five-story and ten-story structural models with a story height of 3 m and a bay width of 3 m were used. The soft story was considered as the story heights of 5 m and 7 m at the bottom, middle and top levels of the structural models. I believe this paper presents a nice effort and is relevant and important for the advancing of this topic. However, there are a few questions and comments from the reviewer.

1. Why were 5 story and 10 story buildings considered only? please explain in detail. For example, why were 15, 20 or 30 story buildings not used?

2. Why were 5 m and 7 m used for soft story height? please explain in detail.

3. In lines 218 and 297, Which study? Please specify.

4. The paper should be carefully checked by a native speaker.

Reviewer 3 Report

Dear authors,

I would like to thanks the authors and please consider the following comments:

1-      Have you control the exception of page 97 of ASCE07-2016 for soft story in modeling:

“Vertical structural irregularities of Types 1a, 1b, and 2 in Table 12.3-2 do not apply where no story drift ratio under design lateral seismic force is greater than 130% of the story drift ratio of the next story above. Torsional effects need not be considered in the calculation of story drifts.”

2-      As the assumption of special moment frame in the manuscript and obligatory control of “weak beam strong column “ in special structures, So your example should have soft story? By controlling this criteria soft story shouldn’t be existed.

3-      In the manuscript only  floor acceleration is considered for NSCs. But as you know and as mentioned in FEMA P-58 for example wall partitions are related to interstory drift. So it is a mixed process. How can you justify your assumption that only acceleration is important?

4-      Simplifed methods as introduced in paper: https://doi.org/10.1016/j.jcsr.2018.04.010 would be mentioned in introduction part.

5-      Damage state of NSCs should be stated based on Hazus–MH 2.

6-      FEMA P-58 methodology can be used for loss estimation of NSCs to clear and upgrade the manuscript.

7-      The nonlinear models are important. May be the nonlinear behavior affects the results. Only linear modeling under ground motion can’t be rational. How can you justify this problem?

8-      In ref No:8, what is the name of the journal: In Proceedings of the Structures? Is the style correct?

Reviewer 4 Report

   The manuscript assumes that the soft layer is located at the bottom (ground), middle layer and top layer of the building model under consideration. Evaluate the inter-story displacement and inter-story displacement ratio to evaluate the structural performance. The floor response spectrum and the amplification effect of NSC on the floor acceleration response are studied to understand the behavior of NSC. However, there are still some problems that need to be modified in this article. The specific suggestions are as follows:

1. In the article, only 5-layer and 10-layer structures are used as reference models, and the number of reference models is too small;

2. The change of floor acceleration in the whole building height in the article is nonlinear, and the data obtained by the code formula is not accurate. Does it affect the experimental structure?

3. For soft-story buildings, what is the basis for selecting the height of 5m and 7m as the soft layer? Are these two heights representative?

4. Figure 6(a) is missing the vertical coordinate units, please add them.

5. The peak of the M10ms5 5th floor FRS spectrum in Fig. 9 is not significantly amplified, which is inconsistent with the conclusion in the text, please explain.

6. In the conclusion part of the article, "the main objective of this study is to study how the position of the soft layer affects the inter-story displacement" does not correspond to the simulation "study the influence of the position of the soft layer on the inter-story displacement";

Round 2

Reviewer 1 Report

Despite the nonlinear behaviour of structural elements were not considered, the revised version of the paper was improved. Hence, I recommend the paper for publication.

Reviewer 3 Report

Dear the authors

The mentioned comments are addressed adequately and the paper can be accepted in the present form